# Measuring voluntary responses in healthcare utilization during the COVID-19 pandemic: Evidence from Taiwan

**Yung-Yu Tsai[1], Tzu-Ting Yang [2] \***

**1** Truman School of Government and Public Affairs, University of Missouri, Columbia, MO, United States of America, **2** Institute of Economics, Academia Sinica, Taipei, Taiwan

☯ These authors contributed equally to this work.

\* ttyang@econ.sinica.edu.tw

**Data Availability Statement:** Data and code used in this paper is available at: https://github.com/yungyutsai/Taiwan_COVID19_Healthcare_Utilization.

## Abstract

Healthcare has been one of the most affected sectors during the coronavirus disease 2019 (COVID-19) pandemic. The utilization of related services for non-COVID-19 diseases fell dramatically following the point at which the virus broke out; however, little is known about whether this observed decline in healthcare use was due to voluntary behaviors or enforced measures. This paper quantifies the spontaneous change in healthcare utilization during the pandemic. We utilize a county-by-week-level dataset from Taiwan's National Health Insurance (NHI) record, covering the entire Taiwanese population, and a difference-in-differences design. Our results indicate that even if there were no human mobility restrictions or supply-side constraints, people voluntarily reduced their demand for healthcare, due to fears of contagion, or COVID-related precautionary behaviors. We find that the number of outpatient visits (inpatient admissions) decreased by 19% (10%) during the pandemic period (February to May 2020). Furthermore, the demand response of healthcare for Influenza-like illness (ILI) was much greater and more persistent than for non-ILI, thereby suggesting that the substantial decline in accessing healthcare was induced by positive public health externality of prevention measures for COVID-19. Finally, we find that the demand for healthcare services did not get back to the pre-pandemic baseline, even when there were no local coronavirus cases for 253 consecutive days (mid-April to December 2020) in Taiwan.

## Introduction

COVID-19 has raged through most countries around the world. As of April 2022, it caused 494 million confirmed cases and 6.16 million deaths [1]. The pandemic has changed many aspects of people's lives and had negative impacts on macroeconomic activities [2–6], household consumption [7, 8], and the labor market [9–11]. Particularly, COVID-19 is a public health crisis, and so healthcare systems have been severely affected during the pandemic. Recent studies have shown that there have been large declines in healthcare utilization for

**Funding:** The author(s) received no specific funding for this work.

**Competing interests:** The authors have declared that no competing interests exist.

non-COVID-19 diseases [12–16]. However, it is not clear whether this observed decline is due to voluntary behaviors or to other inevitable issues, such as government restrictions on mobility [12, 14, 17] or the availability of health resources [18–20].

Measuring the voluntary response in healthcare utilization during the pandemic has important policy implications. First, the pandemic significantly strained the capacity of healthcare systems. To free up medical sources for COVID-19 patients, healthcare providers had to restrict or delay the use of services not related to COVID-19 [21]. Since such restrictions might also impede the use of some essential medical care and negatively affect people's health, it raises a question as to what extent the government or healthcare providers have to do so. Previous studies suggest that people change their behaviors voluntarily to reduce the chance of contracting diseases [22–24]. If the spontaneous response is substantial, the government could achieve a similar outcome by implementing policies with fewer restrictions/costs. Second, compared to policy-induced behavior, voluntary responses could be more persistent. Therefore, understanding voluntary change in healthcare use can help us evaluate the possible impact of disease outbreaks on people's health behaviors after the pandemic.

This paper fills the gap by examining the effect of the COVID-19 outbreak and pandemic responses on the voluntary demand for non-COVID-19 healthcare. We utilize a difference-in-differences (DID) design and a 2014–2020 county-by-week-level dataset from Taiwan's National Health Insurance (NHI), covering the entire population in Taiwan. This study aims to identify the impact of the COVID-19 outbreak on healthcare demand. Given that epidemic diseases (especially influenza) are usually seasonal, simply comparing the change in health utilization before and after the pandemic would not provide a valid estimate of the causal effect of the pandemic. Therefore, we utilize the pattern of health utilization in the years without the pandemic as a comparison group, in order to account for any seasonal pattern associated with the health utilization. The DID method estimates the impact of the intervention (the COVID-19 outbreak) by quantifying the differences in changes in outcomes before and after the intervention across the treated (the year 2020) and untreated units (the year 2014–2019). Specifically, we examine whether healthcare utilization during the pandemic (2020) varied compared with corresponding weeks in previous years (2014–2019), after controlling for the county-specific trend in demand for healthcare (e.g., county-by-week fixed effects and county-by-year fixed effects).

The case of Taiwan is well-suited for this analysis—for three important reasons. First, as of the end of 2020, Taiwan had only experienced seven deaths and 799 COVID-19 cases. The total population of Taiwan is around 23 million, so the country's infection rate is very low. Moreover, no severe local transmissions had occurred in 2020. Fig 1 displays the number of new confirmed cases for each week of 2020. The first confirmed COVID-19 case was announced in the 4th week of 2020 (i.e., the week of January 21st). The dark color represents the number of new local confirmed cases in a given week, and the gray color means the weekly number of new non-local cases. Fewer than 10% of confirmed cases are local ones, and there have been no new confirmed local cases from the 17th to 51st week of 2020 (i.e., April 12th to December 22nd, 2020). So the pandemic had a very limited impact on healthcare capacity in the country, thereby helping us rule out unmet demand due to supply-side factors.

Second, Taiwan did not implement any lockdown or self-isolation policies in 2020; the Taiwanese, for their part, carried on with their normal lives alongside relatively looser regulations. Therefore, our estimated change in demand for healthcare can persuasively represent the voluntary response to the COVID-19 pandemic rather than government restrictions on activity/mobility. Furthermore, Taiwan had a consecutive 253 days of no local cases from April 12th to the end of 2020 [25]. Basically, Taiwanese people returned to normal life during this period. This experience gives us a unique chance to examine whether the behavioral change in

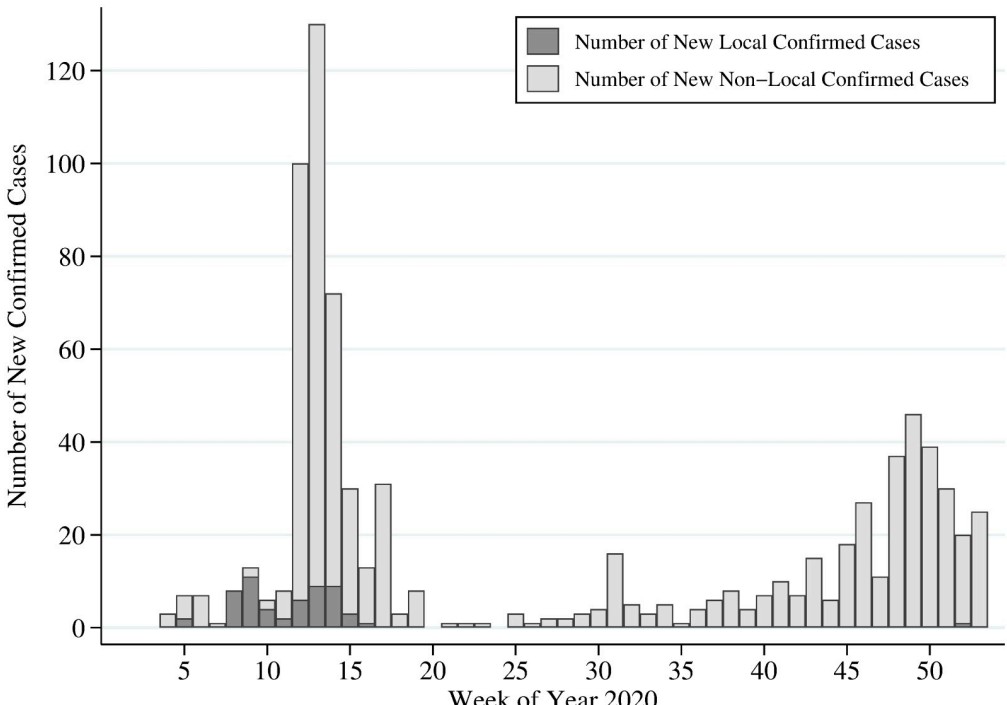

**Fig 1. Weekly number of newly confirmed COVID-19 cases in Taiwan.** This figure displays weekly numbers of newly confirmed COVID-19 cases in Taiwan. The dark color represents the number of new local confirmed cases in a given week, and the gray color means weekly numbers of new imported cases. Data source comes from TCDC.

healthcare utilization can persist in a "COVID-free" period. Though Taiwan experienced a second-wave outbreak of local transmissions from May 15[th] to the end of June 2021, we do not include the 2021 period in our analysis.

Finally, Taiwan's NHI is a compulsory single-payer system in which everyone has to enroll; thus, NHI data cover the population-wide healthcare utilization of both outpatient care and inpatient care. This feature allows us to investigate the pandemic effect on different types of healthcare use. Specifically, in this paper, we identify the impact on both the Influenza-like illness (ILI) diseases and non-ILI diseases.

This paper stands apart from the previous literature in the following ways. First, we contribute to the fast-growing body of literature analyzing the impacts of the COVID-19 pandemic on healthcare systems. Several recent studies indicate that there was a large decline in healthcare utilization during the pandemic period in the US [12–16], UK [26], Spain [27], and Italy [28]. Our paper provides novel evidence by showing the patterns in health utilization when there are no government restrictions on human mobility (e.g., lockdowns or stay-at-home orders) and no supply-side constraints (e.g., inadequate healthcare resources for non-COVID-19 patients).

Second, the spontaneous response to COVID-19 risk broadly relates to the "prevalence response" in the economic epidemiology literature [29–34]. This study is also related to the body of theoretical and empirical literature on voluntary avoidance behavior [35–38]. For example, several recent studies have found that people's voluntary response plays an important role in what decisions they make in terms of mobility, social distancing, and mask-wearing during the pandemic [23, 39–43]. We contribute to this stream of the literature by quantifying the voluntary avoidance of healthcare utilization during the pandemic.

Third, this paper is related to the literature on the health effects of non-pharmaceutical interventions (NPIs). Most of the recent research points out that such interventions can indeed effectively reduce COVID-19 transmission [44–48] and might lead to unintended health benefits [49, 50]. For example, Feng et al. found that COVID-19 outbreaks and corresponding NPIs (e.g. lockdown or shelter-in-place order) substantially reduced influenza activity in China and US [50]. However, Taiwan did not implement such high-cost NPIs against COVID-19. Instead, it used relatively low-cost ones, such as wearing face masks and washing hands and relied on individual distancing efforts. We contribute to this stream of literature by showing whether the interventions based on individual responsibility can still achieve similar health benefits by reducing the incidence of non-COVID-19 diseases.

The primary aim of this paper is to examine the voluntary response in health demand during the COVID-19 outbreak. In addition, this paper also utilizes the unique "COVID-free" period in Taiwan during the second half of 2020 to examine whether and to what extent the spontaneous response persists in the post-pandemic period. Furthermore, this paper compares the heterogeneous impact of outpatient and inpatient care as well as the ILI and non-ILI diseases to decompose the potential mechanisms of the pandemic's impact on health demand responses.

## Materials and methods

### Data

Our healthcare utilization data originates from the Taiwan National ILI Disease Statistics System, accessed via the Taiwan Center for Disease Control's (TCDC) Open Data Portal [51]. This database holds NHI claim data, covering almost the entire population's healthcare utilization. In order to investigate the outbreak of selected infectious diseases in a timely manner, the TCDC provides the public with weekly data on the numbers of outpatient visits and inpatient admissions by county, age group, and category of selected infectious diseases. Note that the definition of "week" in this database follows the World Health Organization (WHO)'s definition, which always begins on a Sunday and ends on a Saturday, but does not definitely start from January 1st.

To construct our outcome variables—the incidence rate of outpatient visits/inpatient admissions per 100,000 population for specific types of diseases—we divide the number of outpatient visits and inpatient admissions by the population of each corresponding county per year. Population information comes from the population statistics database provided by the Department of Household Registration (DHR), Ministry of Interior (MOI), Taiwan [52]. Population data are measured on a monthly basis. We use the month of the end day of each week to link the weekly day on health utilization with the monthly population data.

In our estimated sample, we also include time-varying covariates, such as county-level demographic and weather variables, that may affect health utilization in each county. We acquire demographic data, such as age structure, sex ratio, and educational attainment, from the MOI statistics portal [53]. The above demographic variables are measured on an annual basis. We use the year of the end day of each week to link the weekly day on health utilization with the annual data. We retrieve daily weather information from the Central Weather Bureau's (CWB) observation data inquiry system [54] in order to calculate the weekly average temperature and rainfall for each county.

### Sample

The estimated sample is at the weekly-county level. The sample period is from 2014 to 2020, and we use data from the first to the 52th week. Thus, the estimated sample includes 22

counties × 52 weeks × 7 years (2014 to 2020), leading to a sample size of 8,008. Furthermore, in order to examine the mechanisms behind COVID-19 effects, we categorize diseases into ILI (include influenza, non-COVID-19 pneumonia, and acute upper respiratory infections) and non-ILI.

Table 1 displays summary statistics for the outcome variables and covariates during the pre-outbreak period (i.e., the first three weeks of a year) and the post-outbreak period (i.e., the 4th to 52nd weeks of a year) in the treated year (i.e., 2020) and untreated years (i.e., 2014–2019).

## Empirical specifications

The empirical challenge in identifying the causal effect of the COVID-19 outbreak is that the health utilization changes according to the seasonal patterns and holidays over a year. For example, as shown in Table 1, in the years 2014–2019, the outpatient visits rates for ILI diseases were around 2.7 thousand visits per 100 thousand people in the first three weeks. However, the visit rate dropped to 2.2 thousand visits per 100 thousand people in the fourth week and to the end of the year. This change reflects that the winter season is usually the peak of ILI diseases. Therefore, simply comparing the change in health utilization before and after the COVID-19 outbreak in 2020 might have been confounded by the disease's seasonality.

We apply a difference-in-differences (DID) design to overcome the empirical challenges. Since the first COVID-19 case in Taiwan was reported on January 21st 2020 (i.e., the 4th week of a year), inspired by previous studies [55–57], we use 2020 as the treated year and define the 1st to 3rd weeks and 4th to 52th weeks of the year as the pre-outbreak and post-outbreak periods, respectively. To control for the seasonal pattern of healthcare demand unrelated to the COVID-19 outbreak, we use 2014–2019 as untreated years to construct the counterfactual trend of health utilization in 2020.

We first use a conventional DID design to examine the average effects of the COVID-19 outbreak on healthcare utilization, following which we extend the design to a multiple period DID and an event study design, to investigate further the dynamic trajectory of COVID-19 effects.

**Conventional DID.** In order to obtain average effect of COVID-19 outbreak on healthcare utilization, we estimate the following DID specification:

$$\ln(H_{idt}) = \gamma_0 Y_{2020} \times Post_d + \lambda_i + \eta_d + \theta_t + \lambda_i \times \eta_d$$
$$+ \lambda_i \times \theta_t + X_{idt}\psi + \varepsilon_{idt}. \tag{1}$$

Our estimation is implemented at the weekly-county level. $H_{idt}$ represents the outcomes of interest, namely, numbers of outpatient visits (inpatient admissions) per 100,000 population in county $i$ in week $d$ of year $t$. We focus on three measures of healthcare utilization: 1) visits/admissions for all diseases; 2) visits/admissions for ILI diseases; and 3) visits/admissions for non-ILI diseases. $Y_{2020}$ is a dummy for the treated year that takes one if the observation is in 2020, and zero otherwise. $Post_d$ is a dummy indicating the post-outbreak period (after the 4th week of a year). We include the county fixed effect $\lambda_i$ to control for any time-invariant confounding factors at the county level. The week-of-the-year fixed effect $\eta_d$ controls for seasonal patterns in healthcare utilization at the national level within a year and year fixed effect $\theta_t$ controls for the general trend in healthcare utilization over time. To account for any county-specific seasonal patterns or health shocks, we include the county-by-week fixed effect ($\lambda_i \times \eta_d$) and the county-by-year fixed effect ($\lambda_i \times \theta_t$). $X_{idt}$ refers to a set of covariates, including various holiday dummies (e.g., the Lunar New Year week) and county-level variables, such as age structure, sex ratio, educational attainment, average temperature, average rainfall, and county-specific linear time trend (constructed as the number of weeks from the first week of 2014).

**Table 1. Summary statistics for the treated and untreated years.**

| | Treated Year 2020 | | Untreated Years 2014–2019 | |
|---|---|---|---|---|
| | **Pre-outbreak** | **Post-outbreak** | **Pre-outbreak** | **Post-outbreak** |
| **Outcome Variables** (per 100,000 population) | | | | |
| Number of total outpatient visits | 21,686.64 | 18,713.76 | 20,915.46 | 20,268.92 |
| | (5,458.21) | (4,904.24) | (5,644.77) | (5,649.82) |
| Number of outpatient visits for ILI diseases | 2,745.49 | 1,456.45 | 2,706.18 | 2,182.31 |
| | (532.46) | (495.06) | (562.34) | (615.41) |
| Number of outpatient visits for Non-ILI diseases | 18,941.15 | 17,257.31 | 18,209.29 | 18,086.61 |
| | (5,187.03) | (4,744.46) | (5,360.66) | (5,369.81) |
| Number of total inpatient admissions | 166.86 | 168.52 | 148.20 | 158.38 |
| | (87.62) | (88.12) | (85.92) | (86.30) |
| Number of inpatient admissions for ILI diseases | 13.48 | 8.01 | 10.73 | 11.29 |
| | (7.66) | (4.62) | (6.90) | (6.74) |
| Number of inpatient admissions for Non-ILI diseases | 153.38 | 160.50 | 137.47 | 147.09 |
| | (84.69) | (85.70) | (81.37) | (82.35) |
| Number of ILI deaths | 1.93 | 1.70 | 1.76 | 1.69 |
| | (0.09) | (0.23) | (0.17) | (0.27) |
| **Demographic Variables** | | | | |
| Population Size (1,000) | 1,072.92 | 1,071.82 | 1,068.33 | 1,068.98 |
| | (1,110.47) | (1,102.25) | (1,091.12) | (1,091.35) |
| Share of Age below 14 | 0.12 | 0.12 | 0.13 | 0.13 |
| | (0.02) | (0.02) | (0.02) | (0.02) |
| Share of Age between 15 to 64 | 0.72 | 0.72 | 0.73 | 0.73 |
| | (0.02) | (0.02) | (0.02) | (0.02) |
| Share of Age above 65 | 0.16 | 0.16 | 0.14 | 0.14 |
| | (0.02) | (0.02) | (0.02) | (0.02) |
| Sex Ratio (Female to Male) | 102.61 | 102.61 | 103.25 | 103.25 |
| | (8.45) | (8.39) | (7.90) | (7.89) |
| Share of Post-Secondary Degree | 0.44 | 0.44 | 0.41 | 0.41 |
| | (0.08) | (0.08) | (0.08) | (0.08) |
| Share of High-School Degree | 0.43 | 0.43 | 0.44 | 0.44 |
| | (0.05) | (0.05) | (0.05) | (0.05) |
| Share of Non-High-School Degree | 0.13 | 0.13 | 0.15 | 0.15 |
| | (0.04) | (0.03) | (0.04) | (0.04) |
| **Weather Variables** | | | | |
| Temperature (˚C) | 17.12 | 22.95 | 16.05 | 22.65 |
| | (2.12) | (4.68) | (2.69) | (4.85) |
| Precipitation (mm) | 7.48 | 4.47 | 3.34 | 5.83 |
| | (10.98) | (7.13) | (5.93) | (9.59) |
| Observations | 66 | 1,078 | 396 | 6,468 |

*Note:* This table displays summary statistics for the outcome variables and covariates during the pre-outbreak period (i.e., the first three weeks of a year) and the post-outbreak period (i.e., the 4th to 52nd weeks of a year) in the treated year (i.e., 2020) and untreated years (i.e., 2014–2019). Healthcare utilization data comes from the Taiwan National ILI Disease Statistics System, which originates from 2014–2020 NHI claim data. We divide the number of outpatient visits and inpatient admissions by the population of each corresponding county per month to obtain the incidence rate of outpatient visits/inpatient admissions per 100,000 population for specific types of diseases. Demographic information comes from the population statistics database provided by the Ministry of Interior (MOI), Taiwan. Population size is measured on monthly basis, and other demographic variables are measured on annual basis. Weather variables are from the Central Weather Bureau's (CWB) observation data inquiry system. Standard deviations are in parentheses.

We estimate the Eq (1) using a Poisson regression since the outcome variable is weekly counts of outpatient visits/inpatient admissions. We perform our analysis with Stata 16 through the package *ppmlhdfe* [58].

Throughout our analysis, we use the multiway clustering approach proposed by Cameron, Gelbach, and Miller [59] to calculate standard errors clustered at both the year-week and the county levels. We adopt the multiway clustered standard error due to two concerns. First, there are potential within-group correlations of errors associated with both the county cluster and the year-week cluster. We apply the clustered standard error approach to account for this correlation pattern. Secondly, there is an overdispersion issue in our outcome variables. As shown in Table 1, the standard deviations of the outpatient visits/inpatient admissions are far exceed the squared root of the mean. Overdispersion problems in count data are sometimes caused by clustering [60]. We observe in our data that any variances in outcomes variables within the county cluster and the year-week cluster are smaller than the variances for the whole sample. Therefore, we apply the multiway clustered standard error to account for the overdispersion problems due to clustering. Finally, we conduct robustness checks on the estimates by computing the standard errors at different cluster levels.

Finally, all regressions are weighted by the monthly population size of a county. As shown in Table 1, a fairly variation in population size across counties exists. Specifically, the population size ranges from 12 thousand to 4 million. To obtain estimates that represent the population average, we weighted all of our estimations with population size. In a latter section, we also report the results on unweighted regressions as a robustness check.

The key variable used for identification in the Eq (1) is an interaction term between an indicator for the treated year $Y_{2020}$ and a dummy for the post-outbreak period $Post_d$. The coefficients of interest are $\gamma_0$, measuring the difference in healthcare utilization before and after the COVID-19 outbreak in 2020 (i.e., the treated year), relative to the difference in the corresponding periods for 2014–2019 (i.e., the untreated years). $\gamma_0$ can represent COVID-19 effects on healthcare utilization if the common trend assumption holds. That is, in the absence of the COVID-19 outbreak, the weekly trend in healthcare utilization should be similar in the treated and the untreated years. We examine this assumption by using the DID event study design and a set of placebo tests.

**Multiple period DID.**   In 2020, Taiwan had no local COVID-19 cases from April 12[th] to the end of 2020 (i.e., around 253 days), so the government relaxed preventive measures for COVID-19 on June 7[th] (i.e., the 24[th] week of 2020) and suggested that people could return to a normal life. Therefore, we divide the post-outbreak period into 1) pandemic period and 2) COVID-free period, and estimate the following multi-period DID specification:

$$\ln(H_{idt}) = \quad \gamma_1 Y_{2020} \times Pandemic_d + \gamma_2 Y_{2020} \times CovidFree_d$$
$$+\lambda_i + \eta_d + \theta_t + \lambda_i \times \eta_d + \lambda_i \times \theta_t + X_{idt}\psi + \varepsilon_{idt}. \tag{2}$$

*Pandemic$_d$* denotes the dummy variable, which takes a value of 1 during the 4[th] to the 23[rd] weeks (the pandemic period). *CovidFree$_d$* takes the value 1 during 24[th] to the 52[nd] weeks (COVID-free period). The coefficients of interest are $\gamma_1$ ($\gamma_2$), which measure changes in healthcare utilization during the pandemic period (COVID-free period) compared to the pre-outbreak period (the first three weeks of a year) in 2020, relative to the corresponding weeks in 2014–2019. If the COVID-19 effects faded out when there were no local cases, the $\gamma_2$ should shrink to zero.

**Event study.**   In order to examine common trend assumption and outline the full dynamic trajectory of the COVID-19 effects, we implement an event study design by interacting the

treated year dummy $Y_{2020}$ with lead and lag time dummies $W_d$.

$$\ln(H_{idt}) = \sum_d \beta_d Y_{2020} \times W_d + \lambda_i + \eta_d + \theta_t + \lambda_i \times \eta_d + \lambda_i \times \theta_t + X_{idt}\psi + \varepsilon_{idt}. \quad (3)$$

We use $W_d$, where $d = -3, -1, 0, 1, 2, 3, \ldots.46, 47, 48$, to denote dummy variables for the weeks before and after the 4th week of a year. For example, $W_1$ represents a dummy for the first week after the announcement of the first confirmed COVID-19 case. Note that we use the 2nd week of a year as the baseline week (i.e., $d = -2$).

The key variables used for identification in Eq (3) are a set of week dummies $W_d$ interacted with the treated year dummy $Y_{2020}$. The coefficients of interest are $\beta_d$, measuring the difference in healthcare utilization between week $d$ and the baseline week for 2020 (i.e., the treated year), relative to the difference for 2014–2019 (i.e., the untreated years). $\beta_d$ can represent the dynamic effects of the COVID-19 outbreak on healthcare utilization.

## Results

### Graphical evidence

Fig 2 displays trends in the utilization of outpatient care and inpatient care. The solid line represents the trend in 2020, and the dashed line denotes the average across 2014–2019 with a 95% confidence interval. Since the first confirmed COVID-19 case was announced in the 4th week of 2020, the vertical line in the graph separates that week from the following weeks. The vertical axis in Fig 2 stands for the percentage change in the number of outpatient visits (inpatient admissions) from the baseline week, which is the second week of each year.

Fig 2A shows the trend in outpatient visits for ILI diseases. As the flu season usually runs from October to April in Taiwan, numbers of visits for ILI diseases tend to be higher until April during 2014–2019 (see the dashed line). In a sharp deviation from the usual seasonal patterns of 2014–2019, the numbers of ILI visits fell by 30% to 65% after the COVID-19 outbreak, and the declining pattern was still persistent at the end of 2020 (see the solid line). For non-ILI diseases, Fig 2B shows that there was a large decline in visits for non-ILI diseases during the lunar new year (i.e., the 5th week of 2020) and a rebound after the holiday. This pattern can also be found in 2014–2019. The difference between 2020 and the previous years is that the average numbers of visits for non-ILI diseases in 2020 did not rebound back to the baseline level, and in fact they declined by around 15% until the middle of the year.

Fig 2C illustrates the evolution of inpatient utilization for ILI diseases. Compared to the baseline weeks, admissions for ILI diseases decreased by 15% to 60% after the virus outbreak in 2020, which is very different from the trends in 2014–2019. In contrast, Fig 2D suggests inpatient admissions for non-ILI diseases in 2020 largely followed patterns similar to those in 2014–2019 except that there was a small decline during the 6th week to the 13th week after the first COVID-19 case.

### COVID-19 effects on healthcare utilization

Table 2 reports the DID estimates (i.e., the coefficient on $Y_{2020} \times Post$ of Eq (1)) in the form of incidence rate ratios (IRR). We display the raw estimated coefficients in Table B1 of the S2 File. The first four columns in Table 2 display results for outpatient care. We gradually include covariates to test the sensitivity of the results. Estimates across the specifications are fairly independent of the introduction of different sets of covariates and fixed effects. Our preferred specification is in Column (4) of Table 2, which includes a full set of covariates. The estimate in Column (4) of Panel A suggests that compared to the same weeks in 2014–2019, total outpatient visits during the post-outbreak period in 2020 significantly decreased by 13%

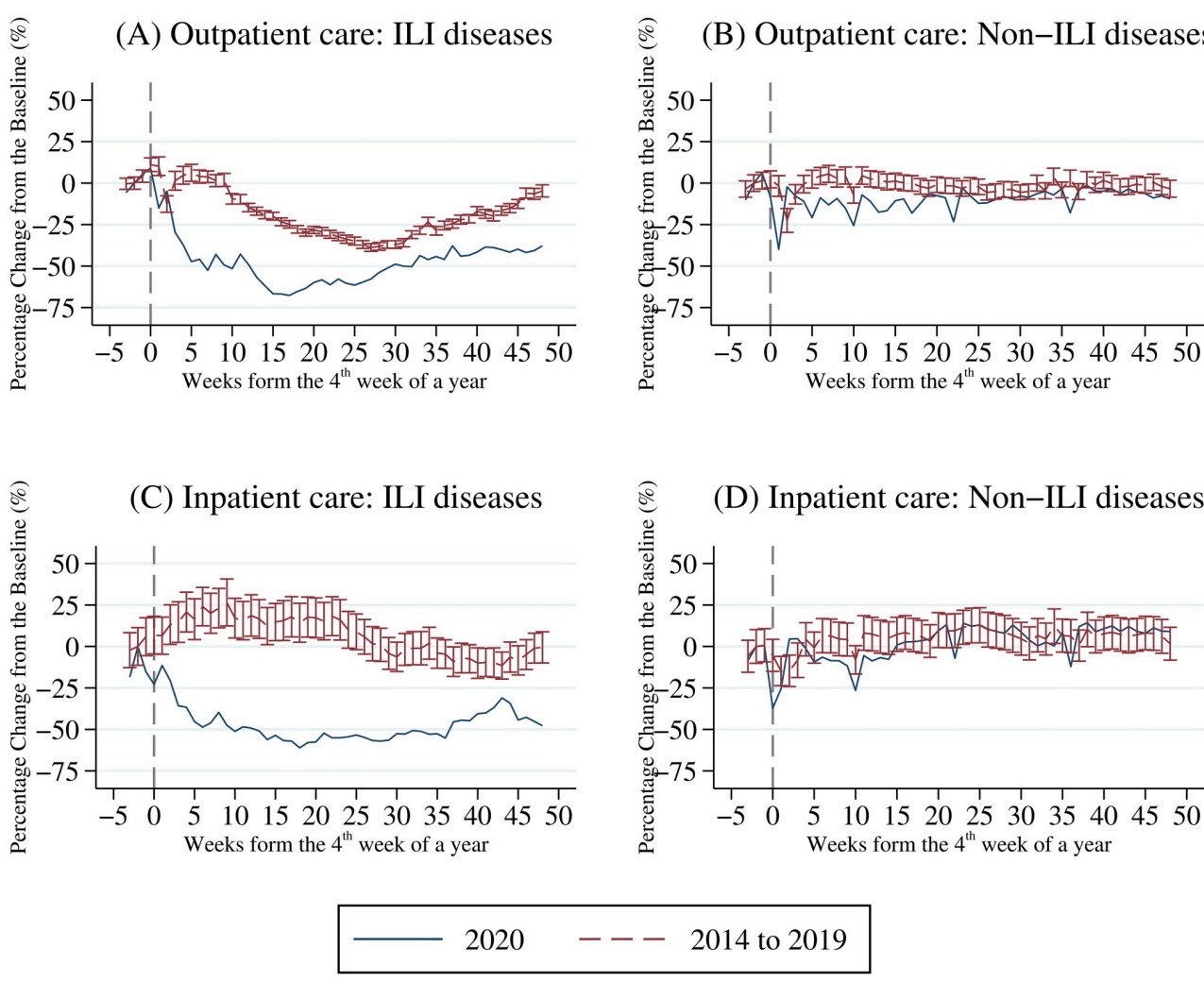

**Fig 2. Percentage change in number of outpatient visits/inpatient admissions.** A: Outpatient care: ILI diseases. B: Outpatient care: Non-ILI diseases. C: Inpatient care: ILI diseases. D: Inpatient care: Non-ILI diseases. Sample period is 2014–2020. The solid line represents the trend in 2020, and the dashed line denotes the average across 2014–2019 with a 95% confidence interval (error bar). Since the first confirmed COVID-19 case was announced in the 4th week of 2020 (i.e. the week of January 21st), the vertical line in the graph separates that week from the following weeks. The vertical axis of this figure stands for the percentage change in the number of outpatient visits (inpatient admissions) from the baseline weeks. We use the number of outpatient visits (inpatient admissions) in the second week of each year as the baseline.

(IRR = 0.87, 95% confidence interval [CI] 0.84 to 0.90). Furthermore, the estimates in Column (4) of Panels B and C show that ILI visits saw a much larger decline (i.e., 38% decrease, IRR = 0.62) than non-ILI visits (i.e., 10% decrease, IRR = 0.9).

Columns (5) to (8) display the DID estimates for inpatient care. Column (8) of Table 2 provides our preferred estimates. The estimate in Column (8) of Panel A indicates that the COVID-19 outbreak reduced total inpatient admissions by 4% (IRR = 0.96, 95% CI 0.90 to 1.01). This estimate is only one-third of the decline in outpatient care. The estimates in Column (8) of Panels B and C show the results for ILI admissions and non-ILI admissions, respectively. Compared to the trend in previous years, the number of ILI admissions declined by 40% (IRR = 0.6, 95% CI 0.54 to 0.68) during the pandemic period in 2020 (see Panel B). In contrast, there was only a negligible decrease in inpatient admissions for non-ILI diseases (see Panel C).

**Table 2. Effects of the COVID-19 outbreak on non-COVID-19 health utilization.**

| | (1) | (2) | (3) | (4) | (5) | (6) | (7) | (8) |
|---|---|---|---|---|---|---|---|---|
| | **Outpatient Care** | | | | **Inpatient Care** | | | |
| **Panel A**: All diseases | | | | | | | | |
| $Y_{2020} \times Post$ | 0.87*** | 0.88*** | 0.87*** | 0.87*** | 0.95*** | 0.96** | 0.95 | 0.96 |
| | [0.86,0.89] | [0.84,0.92] | [0.84,0.90] | [0.84,0.90] | [0.94,0.96] | [0.94,0.99] | [0.90,1.01] | [0.90,1.01] |
| **Panel B**: ILI diseases | | | | | | | | |
| $Y_{2020} \times Post$ | 0.63*** | 0.63*** | 0.62*** | 0.62*** | 0.60*** | 0.60*** | 0.60*** | 0.60*** |
| | [0.59,0.68] | [0.58,0.68] | [0.58,0.67] | [0.57,0.67] | [0.57,0.63] | [0.54,0.65] | [0.53,0.68] | [0.54,0.68] |
| **Panel C**: Non-ILI diseases | | | | | | | | |
| $Y_{2020} \times Post$ | 0.90*** | 0.91*** | 0.90*** | 0.90*** | 0.98*** | 0.99 | 0.98 | 0.98 |
| | [0.89,0.91] | [0.87,0.94] | [0.87,0.93] | [0.87,0.93] | [0.97,0.98] | [0.96,1.02] | [0.92,1.04] | [0.92,1.05] |
| Observation | 8,008 | | | | | | | |
| Basic control | ✓ | ✓ | ✓ | ✓ | ✓ | ✓ | ✓ | ✓ |
| Demographic variables | | ✓ | ✓ | | | ✓ | ✓ | |
| Weather variables | | ✓ | ✓ | ✓ | | ✓ | ✓ | ✓ |
| County fixed effect | | | ✓ | ✓ | | | ✓ | ✓ |
| County-by-year fixed effect | | | | ✓ | | | | ✓ |
| County-by-week fixed effect | | | | ✓ | | | | ✓ |
| County specific time trend | | | | ✓ | | | | ✓ |

*Note:* This table shows the incidence-rate ratios (IRR) for the estimated $\gamma_0$ (i.e. the coefficient on $Y_{2020} \times Post_d$) in the Eq (1), which is a Poisson regression. Sample period is 2014–2020. *Basic Control* includes the year fixed effect, the week fixed effect and various holiday dummies such as, New Year Eve, New Year, Lunar New Year, Peace Memorial Day, Qing-Ming Festival, Labor's Day, and Dragon Boat Festival, Moon Festival, and National Day. *Demographic Variables* includes annually county-level age structure, sex ratio, educational attainment. *Weather Variables* includes weekly county-level temperatures and precipitation. All regressions are weighted by the monthly population size of a county. Robust standard errors clustered at the year-week and county levels. 95% CI reported in squared brackets.

*$p < 0.05$

**$p < 0.01$

***$p < 0.001$

## COVID-19 effects during and after the pandemic

The first four columns of Table 3 illustrate the results for outpatient care. We report the IRR here and include the estimated coefficients in Table B2 in the S2 File. Again, we gradually include covariates to examine the robustness of our estimates to various specifications. The DID estimates in all specifications are fairly stable. The estimate in Column (4) of Panel A suggests that compared to the same weeks in 2014–2019, total outpatient visits during the pandemic period in 2020 significantly decreased by 19% (IRR = 0.81, 95% CI 0.78 to 0.85). In addition, we find that the reduction in outpatient utilization persisted during the COVID-free period, but the estimate shrank to a 9% (IRR = 0.91, 95% CI 0.88 to 0.94) decline. The estimates in Panel B indicate that ILI visits experienced a large decline in both the pandemic period (44% decrease, IRR = 0.56) and the COVID-free period (33% decrease, IRR = 0.67). However, we find that outpatient visits for non-ILI diseases saw a 15% drop (IRR = 0.85) during the pandemic period and rebounded to a 6% reduction (IRR = 0.94) during the COVID-free period (see Panel C).

The last four columns of Table 3 show the results for inpatient care. Column (8) of Table 3 is our preferred specification. Panel A indicates that the COVID-19 outbreak did indeed significantly reduce total inpatient admissions by 10% (IRR = 0.9, 95% CI 0.84 to 0.95) in the pandemic period.

**Table 3. Effects of COVID-19 outbreak on health utilization (by pandemic periods).**

| | (1) | (2) | (3) | (4) | (5) | (6) | (7) | (8) |
|---|---|---|---|---|---|---|---|---|
| | **Outpatient Care** | | | | **Inpatient Care** | | | |
| **Panel A**: All diseases | | | | | | | | |
| $Y_{2020} \times Pandemic$ | 0.81*** | 0.82*** | 0.81*** | 0.81*** | 0.89*** | 0.90*** | 0.89*** | 0.90*** |
| | [0.79,0.83] | [0.79,0.86] | [0.78,0.85] | [0.78,0.85] | [0.88,0.89] | [0.88,0.93] | [0.84,0.95] | [0.84,0.95] |
| $Y_{2020} \times CovidFree$ | 0.92*** | 0.92*** | 0.91*** | 0.91*** | 0.99** | 1.00 | 0.99 | 1.00 |
| | [0.90,0.93] | [0.89,0.95] | [0.88,0.94] | [0.88,0.94] | [0.98,1.00] | [0.98,1.03] | [0.93,1.05] | [0.94,1.06] |
| **Panel B**: ILI diseases | | | | | | | | |
| $Y_{2020} \times Pandemic$ | 0.57*** | 0.57*** | 0.57*** | 0.56*** | 0.59*** | 0.58*** | 0.59*** | 0.59*** |
| | [0.51,0.65] | [0.51,0.64] | [0.50,0.64] | [0.50,0.64] | [0.54,0.64] | [0.52,0.66] | [0.50,0.68] | [0.51,0.69] |
| $Y_{2020} \times CovidFree$ | 0.68*** | 0.68*** | 0.67*** | 0.67*** | 0.61*** | 0.60*** | 0.61*** | 0.61*** |
| | [0.64,0.73] | [0.63,0.72] | [0.63,0.72] | [0.62,0.72] | [0.58,0.64] | [0.55,0.66] | [0.54,0.69] | [0.54,0.69] |
| **Panel C**: Non-ILI diseases | | | | | | | | |
| $Y_{2020} \times Pandemic$ | 0.84*** | 0.86*** | 0.85*** | 0.85*** | 0.91*** | 0.93*** | 0.92** | 0.92** |
| | [0.83,0.86] | [0.82,0.89] | [0.81,0.88] | [0.81,0.88] | [0.91,0.91] | [0.90,0.96] | [0.86,0.97] | [0.86,0.98] |
| $Y_{2020} \times CovidFree$ | 0.94*** | 0.95** | 0.94*** | 0.94*** | 1.02*** | 1.03 | 1.02 | 1.02 |
| | [0.93,0.95] | [0.91,0.98] | [0.90,0.97] | [0.90,0.97] | [1.01,1.02] | [1.00,1.06] | [0.96,1.09] | [0.96,1.09] |
| Observation | 8,008 | | | | | | | |
| Basic control | ✓ | ✓ | ✓ | ✓ | ✓ | ✓ | ✓ | ✓ |
| Demographic variables | | ✓ | ✓ | | | ✓ | ✓ | |
| Weather variables | | ✓ | ✓ | ✓ | | ✓ | ✓ | ✓ |
| County fixed effect | | | ✓ | ✓ | | | ✓ | ✓ |
| County-by-year fixed effect | | | | ✓ | | | | ✓ |
| County-by-week fixed effect | | | | ✓ | | | | ✓ |
| County specific time trend | | | | ✓ | | | | ✓ |

*Note:* This table shows the incidence-rate ratios (IRR) for the estimated $\gamma_1$ (i.e. the coefficient on $Y_{2020} \times Pandemic_d$) and $\gamma_2$ (i.e. the coefficient on $Y_{2020} \times CovidFree_d$) in the Eq (2), which is a Poisson regression. Sample period is 2014–2020. *Basic Control* includes the year fixed effect, the week fixed effect and various holiday dummies such as, New Year Eve, New Year, Lunar New Year, Peace Memorial Day, Qing-Ming Festival, Labor's Day, and Dragon Boat Festival, Moon Festival, and National Day. *Demographic Variables* includes annually county-level age structure, sex ratio, educational attainment. *Weather Variables* includes weekly county-level temperatures and precipitation. All regressions are weighted by the monthly population size of a county. Robust standard errors clustered at the year-week and county levels. 95% CI reported in squared brackets.

*$p < 0.05$

**$p < 0.01$

***$p < 0.001$

In contrast, the COVID-19 impact disappeared during the period when Taiwan had no local COVID-19 cases (IRR is close to 1 and statistically insignificant). Interestingly, the estimates in Panel B suggest that the decline in inpatient utilization for ILI diseases continued during both the pandemic period (41% decrease, IRR = 0.59) and the COVID-free period (39% decrease, IRR = 0.61). Similar to the results in Panel A, inpatient admissions for non-ILI diseased only declined during the pandemic period (8% decline, IRR = 0.92) but rebounded to pre-pandemic level in the period without local virus cases.

## Dynamic effects of COVID-19 on healthcare utilization

Fig 3 highlights the IRR of estimated $\beta_d$ in Eq (3), which measures the dynamic effect of the COVID-19 outbreak on healthcare utilization, and the corresponding 95% confidence intervals. We displays the estimated coefficients of the event study model in Fig B1 in the S2 File. In

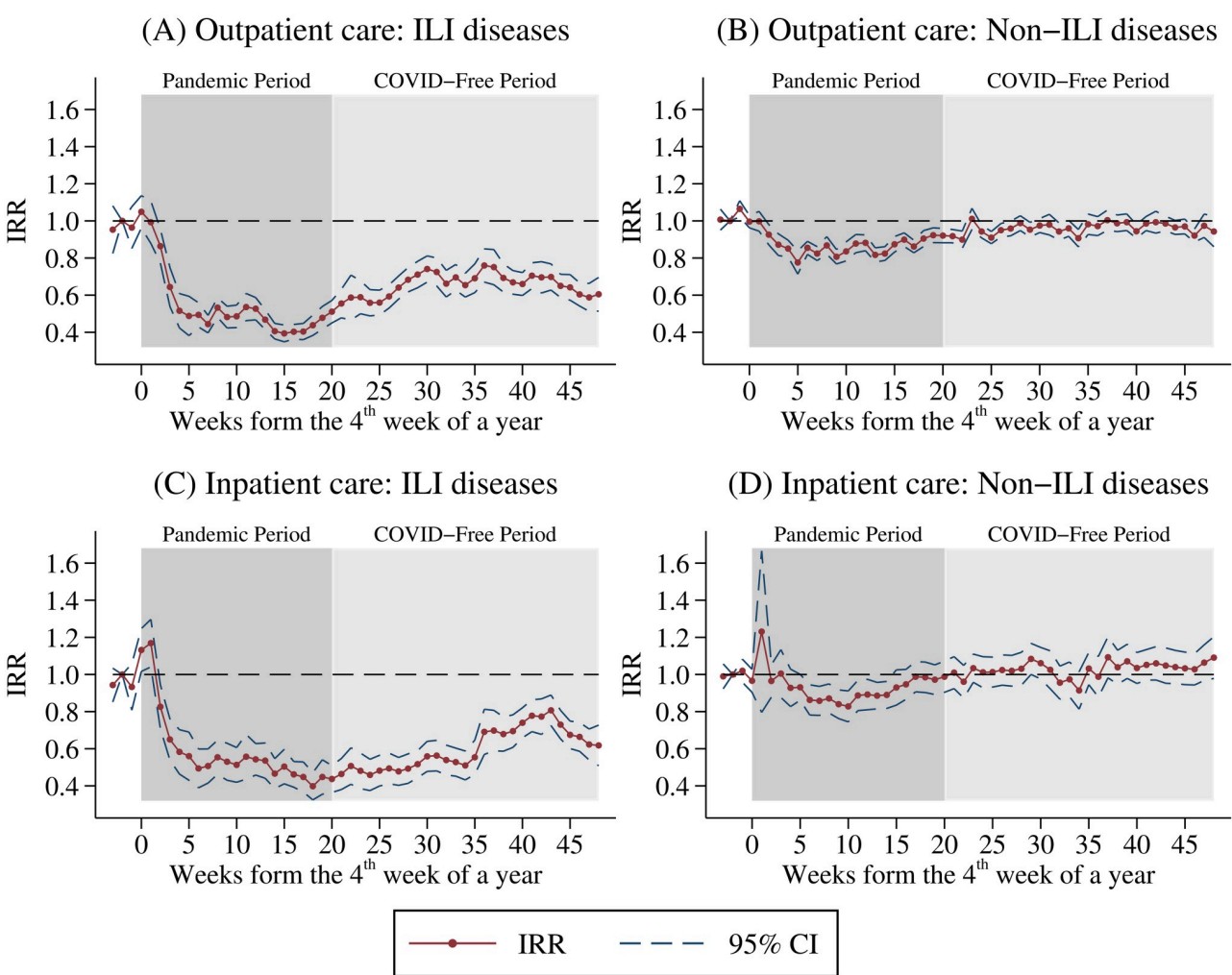

**Fig 3. Dynamic effects of the COVID-19 outbreak on non-COVID-19 health utilization.** A: Outpatient care: ILI diseases. B: Outpatient care: Non-ILI diseases. C: Inpatient care: ILI diseases. D: Inpatient care: Non-ILI diseases. This figure shows the IRR of the estimated $\beta_d$ in Eq (3). The dashed lines represent the corresponding 95% confidence intervals. The horizontal axis denotes the number of weeks from the COVID-19 outbreak (i.e., the 4th week in a year). The top (bottom) panel displays the results for outpatient (inpatient) care by disease type. Sample size is 8,008, and the sample period is 2014–2020.

Fig 3, the horizontal axis denotes the number of weeks from the COVID-19 outbreak (i.e., the 4th week in a year). The top (bottom) panel of Fig 3 displays the results for outpatient (inpatient) care.

Three key insights emerge from the figures. First, estimated IRR for the first three weeks of a year (i.e., $d = -3, -2, -1$) in all figures are close to one, suggesting that trends in the numbers of outpatient visits/inpatient admissions between the treated year (i.e., 2020) and the untreated years (i.e., 2014–2019) were in parallel before the COVID-19 outbreak. Therefore, the common trend assumption of our DID design is valid.

Second, Fig 3A indicates that the size of the reduction in visits for ILI diseases is very large. The COVID-19 outbreak reduced the utilization of outpatient care for ILI diseases by about 50% (IRR is around 0.5) within the first four weeks of the pandemic, and these effects then persisted, thereby suggesting that ILI disease visits still declined by at least 40% (IRR is around 0.6) at the end of the sample period. For visits in relation to non-ILI diseases, Fig 3B suggests

that the reduction in outpatient use is relatively smaller. The number of visits for non-ILI diseases declined by 20% (IRR is around 0.8) in the 4th week after the first case was reported and rebound to pre-pandemic level in late June (i.e., the 23rd week after first COVID-19 case), because there were no local COVID-19 cases in Taiwan for around two months.

Third, Fig 3C suggests that the number of inpatient admissions for ILI diseases decreased by about 40% (IRR = 0.6) in the 4th week after the announcement of the first COVID-19 case. Consistent with outpatient care, the reduction in ILI diseases admissions never rebounded to the pre-pandemic level until the end of 2020. Fig 3D indicates that inpatient admissions for non-ILI diseases significantly declined by nearly 20% (IRR is around 0.8) when the number of COVID-19 cases accumulated quickly and reached peak (i.e., the 6th week to the 13th week after the first COVID-19 case). Interestingly, the COVID-19-induced decline in the number of admissions for non-ILI diseases dropped immediately to zero when Taiwan began to have no local case (i.e., the 14th week after the first COVID-19 case).

## Placebo test and robustness checks

In this section, we first implement a series of placebo tests by excluding observations in 2020 and only using the 2014–2019 sample. Following previous studies [56, 61, 62], we randomly select one year as the pseudo "treated year" for each county and estimate Eq (1). We repeat the above procedures 1,000 times to obtain the distribution of placebo estimates. Fig 4 compares the real estimate with these placebo estimates. Our results suggest that for the outpatient care of all diseases and inpatient care of ILI diseases, the real estimates are way below the placebo ones (see Fig 4A–4C). In sum, this placebo test indicates that the significant estimates in Table 2 should be treated as causal and are not just findings made by chance.

We perform the same placebo tests for our multi-period DID design (Eq (2)) and event-study analysis (Eq (3)). Fig 5 displays the results for Eq (2). The red and blue dashed lines denote the real estimates for COVID-19 effects during the pandemic period and COVID-free period, respectively. The placebo test verifies that all significant estimates in Table 3 are not the result of randomness. Fig 6 illustrates the results for Eq (3). The red lines show real estimates, and the gray lines denote 1,000 placebo ones. Again, the falsification test confirms that significant estimates in the event-study analysis are unlikely to be chance findings.

Next, in the S2 File, we conduct several robustness checks, using various specifications. First, we calculate standard errors based on different clustering levels, in order to examine the robustness of statistical inference. In our main specification, we cluster the standard error at both the year-week and the county levels. We also conduct statistical hypothesis tests using standard errors clustered on the county or year-week levels, respectively (see Tables B3 and B4 in S2 File). We find that the statistical significance of the estimates is robust to the standard errors clustered at different levels. Second, Tables B5 and B6 in S2 File show the estimates based on unweighted Poisson regression (Eqs (1) and (2)). Our main results are robust to this change.

## Discussion

### Interpretation of the results

So far, we have found that the COVID-19 outbreak was associated with a substantial reduction in both outpatient and inpatient utilization. In addition, the healthcare utilization for ILI diseases experienced a much larger decline than for non-ILI diseases. Further, negative impacts peaked during the pandemic period, and then began to rebound when there were no local cases. Since COVID-19 had limited impacts on Taiwan's healthcare system, and the government did not implement any mobility-restricted policy or close health facilities, the reduction

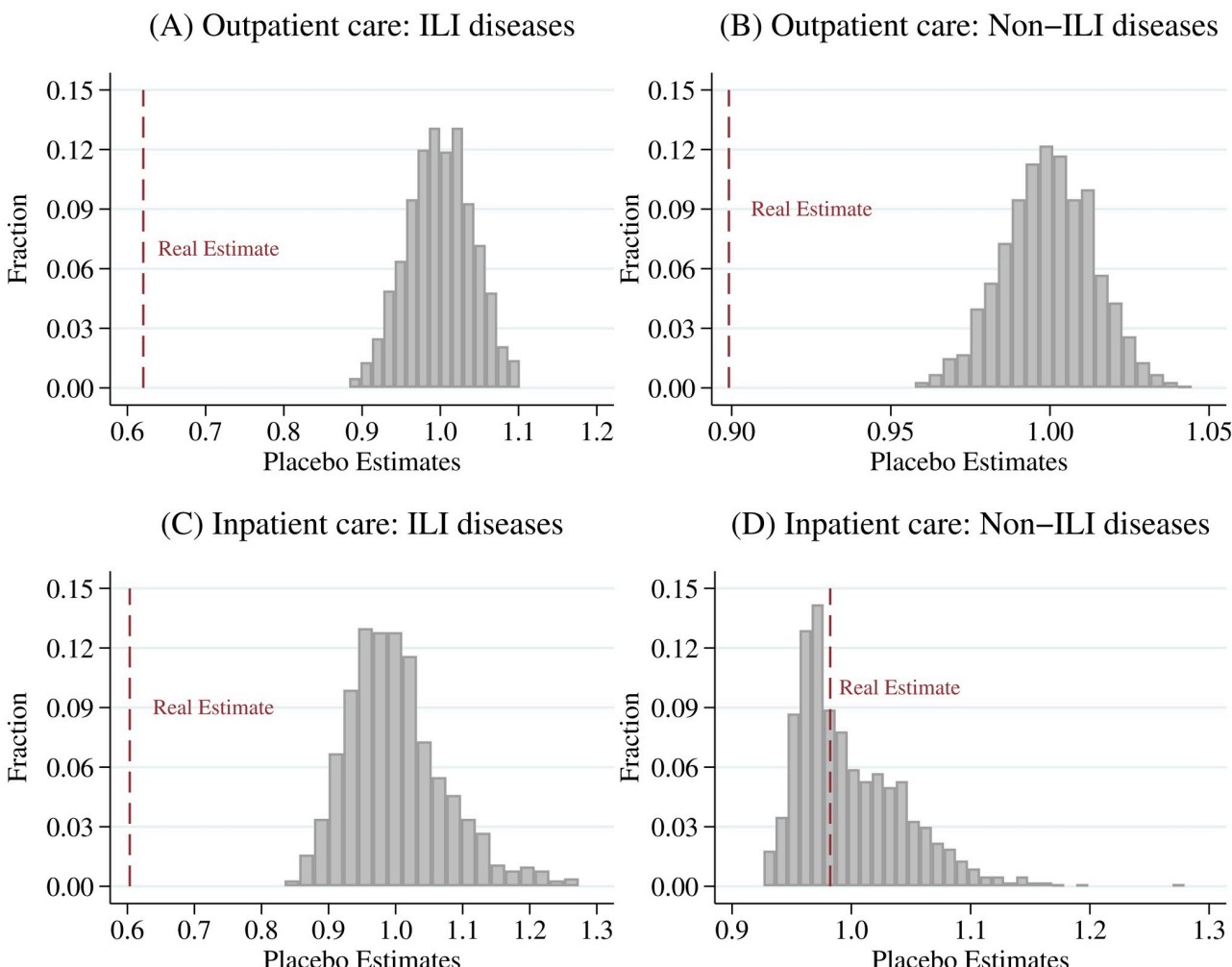

**Fig 4. Placebo test: DID design.** A: Outpatient care: ILI diseases. B: Outpatient care: Non-ILI diseases. C: Inpatient care: ILI diseases. D: Inpatient care: Non-ILI diseases. We use the 2014–2019 sample and randomly select one year as the pseudo "treated year" in each county and estimate Eq (1). We repeat the above procedures 1,000 times to obtain the distribution of placebo estimates. This figure compares our real estimate with these placebo estimates. The vertical axis displays the relative frequency of the estimates. The gray bars denote the placebo estimates, and the red dashed lines denote the real ones.

in healthcare utilization is unlikely to have been caused by issues related to human mobility restrictions or healthcare supply.

Therefore, the decline in demand for healthcare during the pandemic period is likely to be voluntary responses, which are mixed with two effects, namely the fear effect and the prevention effect. First, people may have reduced healthcare utilization due to the fear of COVID-19 infection. As hospitals are usually a place with a high risk of contracting diseases, patients might postpone or cancel their visits if receiving medical treatment is neither necessary nor urgent: we call this mechanism the "fear effect." Second, since ILI diseases (e.g., flu or other forms of pneumonia) shares many similarities with COVID-19 in terms of disease presentation and the ways of transmission, COVID-19 prevention measures, such as wearing a face mask, hand-washing, and social distancing, might have had an unintended effect by reducing the transmission of ILI diseases. Thus, the demand for healthcare could have decreased due to an improvement in health status. This particular mechanism is called the "prevention effect."

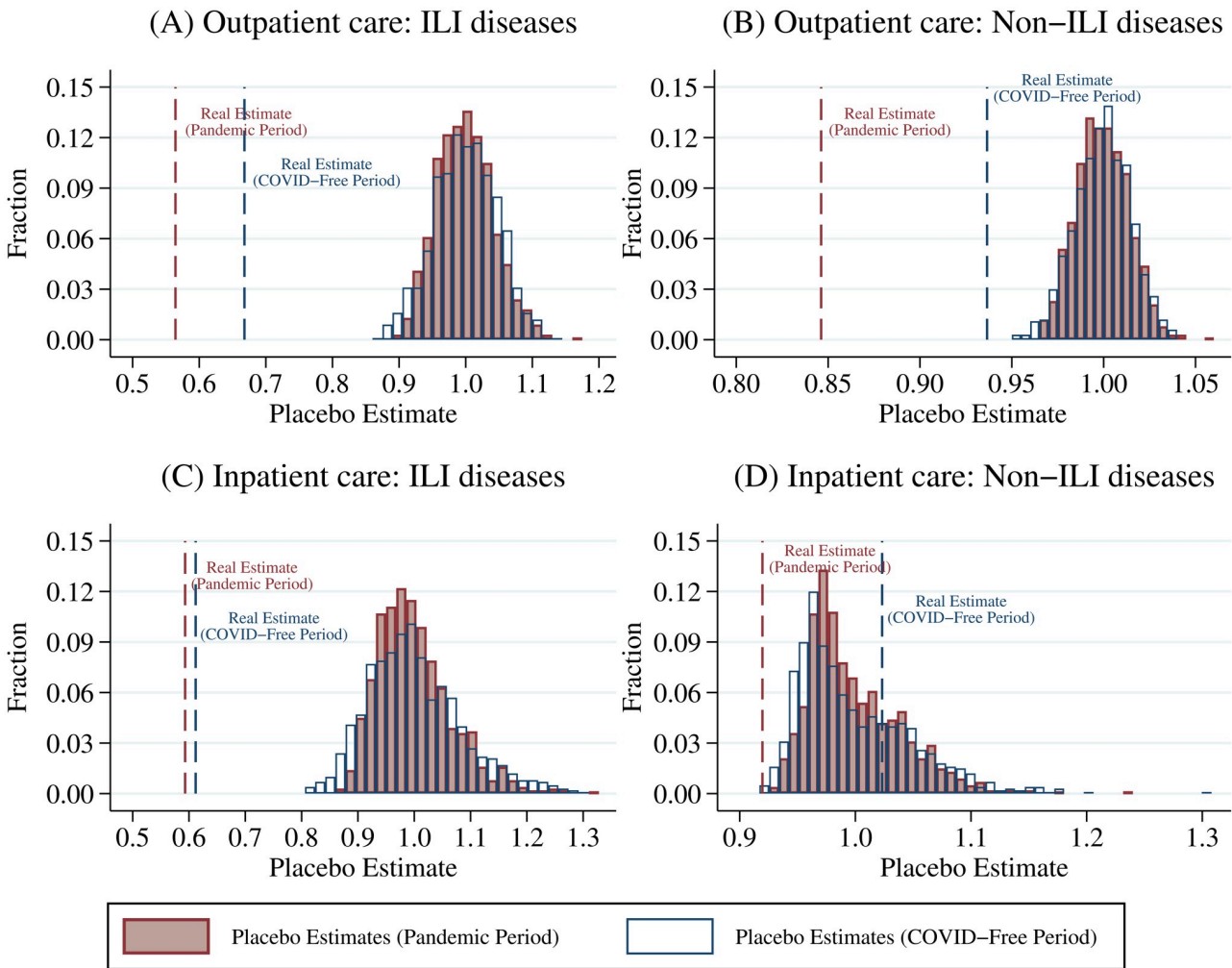

**Fig 5. Placebo test: Multi-period DID design.** A: Outpatient care: ILI diseases. B: Outpatient care: Non-ILI diseases. C: Inpatient care: ILI diseases. D: Inpatient care: Non-ILI diseases. We use the 2014–2019 sample and randomly select one year as the pseudo "treated year" in each county and estimate Eq (2). We repeat the above procedures 1,000 times to obtain the distribution of placebo estimates. This figure compares our real estimate with these placebo estimates. The vertical axis displays the relative frequency of the estimates. The red (blue) dashed line denotes the real estimates for COVID-19 effects during the pandemic period (COVID-free period). The red (blue) bars denote the placebo estimates for COVID-19 effects during the pandemic period (COVID-free period).

In our main analysis, we find that the COVID-19 outbreak caused different impacts on healthcare utilization for ILI diseases and non-ILI diseases. Furthermore, the negative impact of the COVID-19 outbreak was most severe during the pandemic period and faded out when there was very little risk of the local spread of COVID-19 in Taiwan. Such differences help us understand the mechanisms behind healthcare demand responses to the COVID-19 outbreak. If the decline in healthcare utilization were mainly driven by the fear effect, we should expect that the COVID-19 outbreak would have had less of a negative impact on the utilization of inpatient care (i.e., more essential healthcare) than of outpatient care (i.e., less essential health-care). In addition, the demand response to the fear of contracting COVID-19 might have disappeared when the risk of catching the virus was low.

Our results indicate that the demand response of healthcare for non-ILI diseases could have been induced by the fear effect of contracting COVID-19. For non-ILI diseases, the COVID-19 outbreak led to a 13% reduction in outpatient visits but almost no impact on inpatient

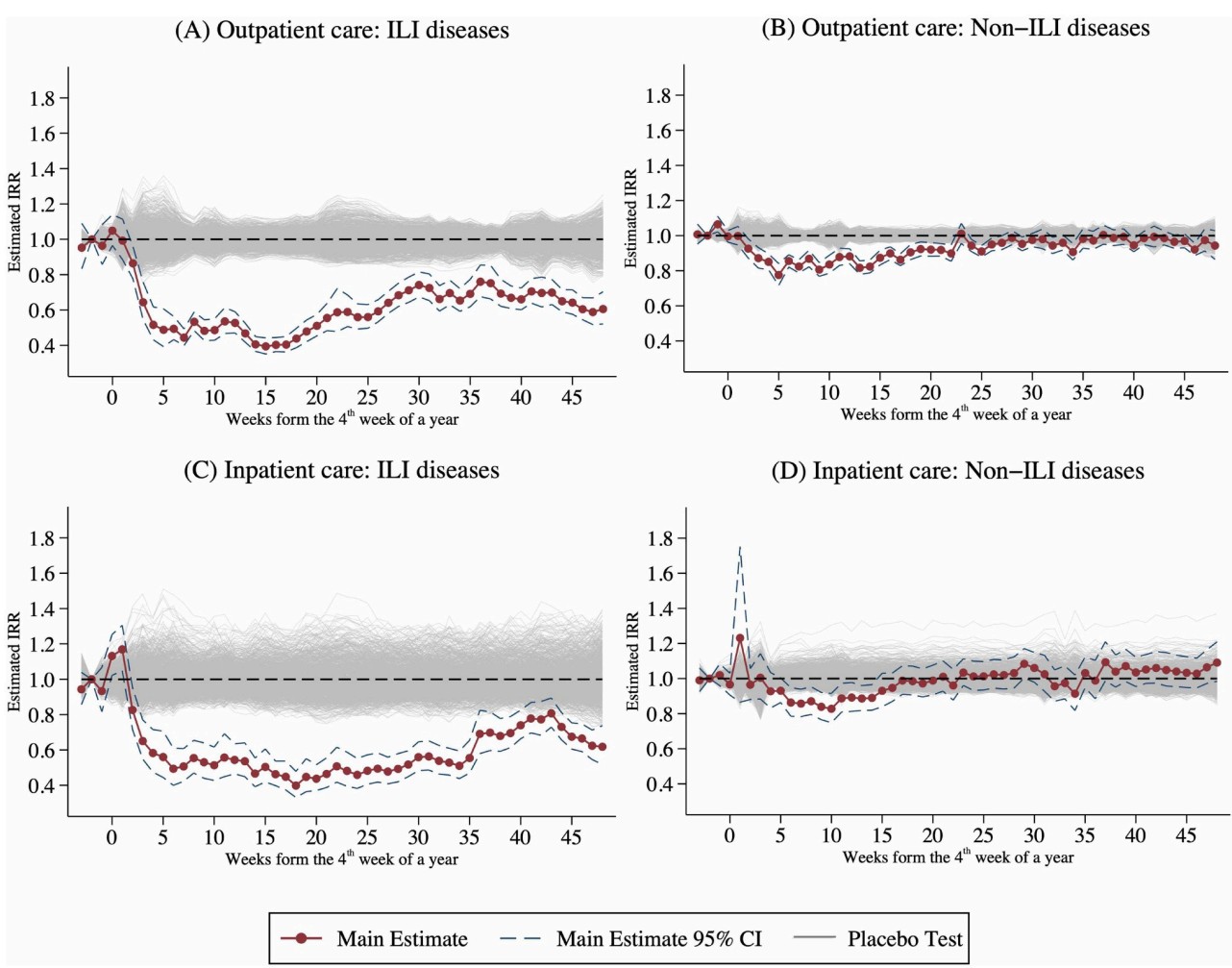

**Fig 6. Placebo test: Event-study design.** A: Outpatient care: ILI diseases. B: Outpatient care: Non-ILI diseases. C: Inpatient care: ILI diseases. D: Inpatient care: Non-ILI diseases. We use the 2014–2019 sample and randomly select one year as the pseudo "treated year" in each county and estimate Eq (3). We repeat the above procedures 1,000 times to obtain the distribution of placebo estimates. This figure compares our real estimates with these placebo estimates. The gray lines denote the placebo estimates, and the red dots and blue dashed lines denote the real estimates and the corresponding 95% confidence intervals.

admissions (i.e., insignificant 4% decrease). Furthermore, we find that the negative effect of the COVID-19 outbreak on the utilization of both outpatient care and inpatient care for non-ILI diseases faded out when no new local COVID-19 cases were reported in Taiwan. Interestingly, the event study analysis indicates that the demand response of inpatient care vanished earlier than for outpatient care (i.e., the 14th week vs. the 23rd week after the first COVID-19 case).

On the other hand, if the reduction in healthcare demand was mainly driven by the effect of COIVD-19 preventive measures, we should expect that the utilization of both outpatient and inpatient care declined similarly because these measures basically helped stop the spread of ILI diseases. In addition, the negative effect should be sizable and persistent since Taiwanese people maintain these healthy habits during the year 2020. For example, Fig A4 of the S1 File shows that the proportion of people who wear a face mask in public spaces remained at over 80% throughout the whole of 2020.

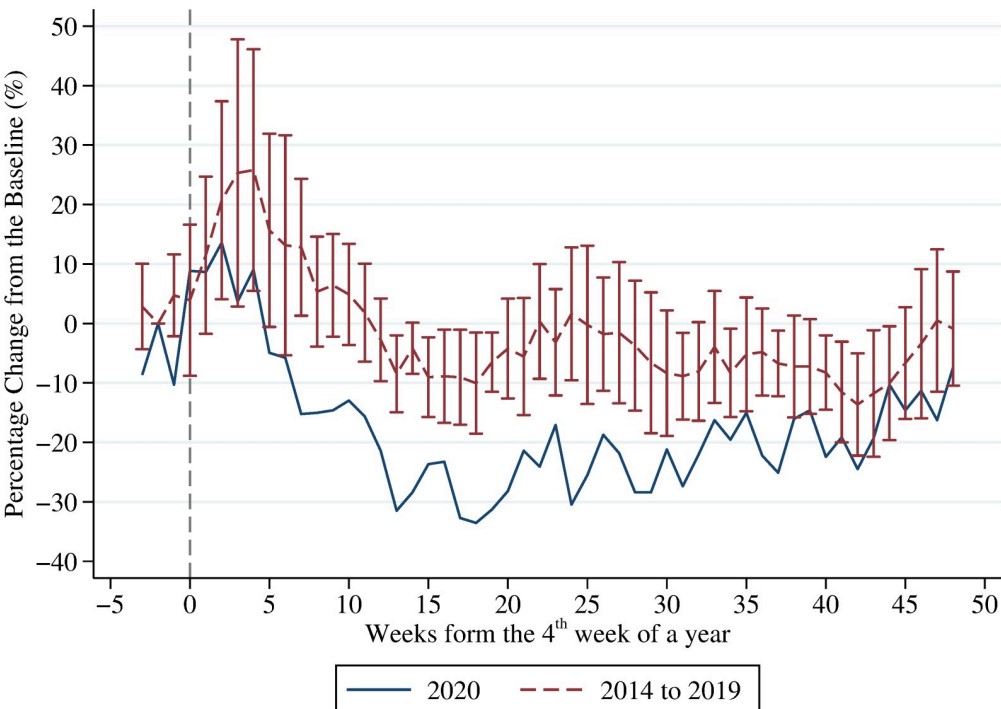

**Fig 7. Effects of COVID-19 outbreak on ILI mortality.** This figure displays the percentage change in the weekly number of ILI-related deaths per 100,000 population from the baseline mean (i.e., the average outcome of the second weeks in each year). The solid line represents the trend of the ILI-related mortality rate in 2020. The dashed line represents the 6-year average of the ILI-related mortality rate during 2014–2019 and the corresponding 95% confidence interval (error bar). The vertical line in the graph denotes the 4th week of a year.

We do find that there were large declines in both outpatient visits and inpatient admissions for ILI diseases after COVID-19 outbreak and the negative effects continued even during the period when Taiwan had no local COVID-19 case. Moreover, if precautions against coronavirus indeed led to a decline in ILI-related healthcare use, we should also find corresponding evidence on improvements in health status, such as a decline in ILI mortality.

Fig 7 displays the percentage changes in the weekly number of ILI-related deaths per 100,000 population from the baseline mean (i.e., the average outcome of the second weeks in each year). The solid line represents the trend of ILI-related mortality rate in 2020. The dashed line represents the 6-year average of the outcome during 2014–2019 and the corresponding 95% confidence interval. Again, the vertical line in the graph denotes the 4th week of a year. Our results suggest that the trend for the ILI-related mortality rate in 2020 is an outlier compared to the same period in the previous 6 years. The percentage change from the baseline mean for the ILI-related mortality rate fell by 10% to 30% after the COVID-19 outbreak. This result is consistent with the findings in the recent literature on the unintended health benefits of NPIs against COVID-19 [49, 50].

## Implications for the observed decline in healthcare utilization

Our results indicate that outpatient visits and inpatient admissions fell by 19% and 10%, respectively, during the pandemic period. For ILI diseases, the declines in outpatient visits and inpatient admissions were even larger (i.e. more than 40% decrease). Compared to estimates in recent studies, we find that the voluntary response is substantial. For example, Birkmeyer

et al. found that hospital admissions for non-COVID-19 diseases decreased by more than 20% from February to April 2020 in the US [13]. In addition, their results suggest that admissions for ILI diseases, such as urinary tract infection and pneumonia, fell by 40% to 50%.

Given the low risk of contracting COVID-19 in Taiwan, we believe our estimates could serve as a "lower bound" for voluntary healthcare utilization responses in other countries. This implies that we may treat voluntary behavior as a major reason for the observed decline in healthcare utilization. Furthermore, our results indicate that the demand for healthcare services did not get back to the pre-pandemic baseline, even after the pandemic died away in Taiwan, thereby suggesting that the COVID-19 outbreak might have had (and be having) a long-term impact on people's health behaviors.

### Implications for healthcare expenditure

Using our results, we can provide the estimated effect of the COVID-19 outbreak on NHI healthcare expenditure. Note that the average expense per outpatient visit and per inpatient admission are around 1,387 NT$ (i.e., 49.5 US$) and 63,249 NT$ (i.e., 2,258.8 US$), respectively. Based on the above information, our DID estimates from column (4) and (8) in Panel A of Table 3 suggest that the COVID-19 outbreak could have "saved" the NHI around 52.4 billion NT$ (i.e., 1.9 billion US$), which accounts for 7.3% of the annual NHI budget.

### Conclusion

This paper examines the effects of the COVID-19 outbreak on voluntary demand for healthcare in Taiwan. By comparing with the same period in the years before the COVID-19 outbreak (i.e. 2014–2019), we find a large decline in health utilization after the outbreak of COVID-19. On average, the number of outpatient visits and inpatient admissions decreased by 19% and 10% during the pandemic period, respectively. Even in the period with no local coronavirus cases, outpatient visits still remains a 9% reduction. In addition, the demand response of healthcare for ILI diseases was much larger and persistent than for non-ILI diseases. Finally, we also find that the outbreak induced a 10% to 30% decline in the ILI-mortality rate.

There are two important limitations of this paper. First, our data source only provides total outpatient visits/inpatient admissions and selected infectious diseases. Due to this limitation, we cannot analyze the impact of COVID-19 on other types of healthcare, such as mental health [63]. It is possible that future researches will explore such issues after individual-level NHI claim data have been released. Second, our research design cannot untangle which prevention measures avoided the spread of ILI diseases and then reduced demand for associated healthcare. However, as universal masking has been the key element in Taiwan's pandemic response, we believe that this practice should play an important role in this regard. Nevertheless, more studies are needed to understand the effect of different prevention measures on health utilization and health.

### Supporting information

**S1 File. Appendix A: Taiwan's response to the COVID-19 pandemic.**
(ZIP)

**S2 File. Appendix B: Additional results.**
(ZIP)

## Acknowledgments

We thank participants of the COVID-19 study group at the Institute of Economics, Academia Sinica for their valuable comments.

## Author Contributions

**Conceptualization:** Yung-Yu Tsai, Tzu-Ting Yang.

**Data curation:** Yung-Yu Tsai.

**Formal analysis:** Yung-Yu Tsai, Tzu-Ting Yang.

**Investigation:** Yung-Yu Tsai, Tzu-Ting Yang.

**Methodology:** Tzu-Ting Yang.

**Project administration:** Tzu-Ting Yang.

**Software:** Tzu-Ting Yang.

**Visualization:** Yung-Yu Tsai.

**Writing – original draft:** Yung-Yu Tsai, Tzu-Ting Yang.

**Writing – review & editing:** Tzu-Ting Yang.

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
