## [Decision Letter · Decision Letter 0]

28 Feb 2022

PONE-D-21-39908Measuring Voluntary Responses in Healthcare Utilization During the COVID-19 Pandemic: Evidence from TaiwanPLOS ONE

Dear Dr. Yang,

 Thank you for submitting your manuscript to PLOS ONE. The reviewers have commented on your manuscript and they are largely positive.  However they have some major comments/questions/concerns that need your attention before this can be considered further. Therefore, we invite you to submit a revised version of the manuscript that addresses the points raised during the review process.

We look forward to receiving your revised manuscript.

Kind regards,

M. Sohel Rahman, Ph.D.

Academic Editor

PLOS ONE

Journal Requirements:

“No”

“NO authors have competing interests”

Additional Editor Comments :

Please carefully attend to the comments of the reviewers and answer the questions raised by Reviewer 2.

Reviewers' comments:

Reviewer's Responses to Questions

**Comments to the Author**

1. Is the manuscript technically sound, and do the data support the conclusions?

Reviewer #1: Partly

Reviewer #2: Yes

2. Has the statistical analysis been performed appropriately and rigorously? 

Reviewer #1: N/A

Reviewer #2: Yes

3. Have the authors made all data underlying the findings in their manuscript fully available?

Reviewer #1: No

Reviewer #2: No

4. Is the manuscript presented in an intelligible fashion and written in standard English?

Reviewer #1: Yes

Reviewer #2: Yes

5. Review Comments to the Author

Reviewer #1: The manuscript proposes some interesting findings on patient's behaviour during COVID-19 pandemic. But i have few suggestions/questions.

1. In Introduction Section, this line should be properly referenced, "Previous studies suggest that people change their behaviors voluntarily to reduce the chance of contracting diseases."

2. In Introduction Section, there is a line " We utilize a difference-in-differences design and a 2014–2020 county-by-week-level dataset from Taiwan’s National Health Insurance (NHI), covering the entire population in Taiwan.". A little description on DID and why DID is appropriate for this study should be mentioned clearly.

3. In Table 3, some of the values are missing, if the values are zeros, they should be mentioned properly.

4. Why are there different p-values were used for different tests? If authors has specific reason to do so, it should be properly mentioned with reference.

5.For mortality analysis, authors used dataset from 2008-2019 but the full analysis was done on 2014-2020. Mortality analysis also needs to be done in the same period to do comparative analysis and draw any conclusions and the corresponding discussions should be updated accordingly.

Reviewer #2: This is an important and relevant report on the assessment of health consequences attributable to the COVID-19 pandemic in Taiwan. The authors do provide a thorough background of the COVID-19 response measures during the pandemic showing a unique position Taiwan (being low COVID-19 burden possibly (i) fear and voluntary concerns, and (ii) the use of masks could be more important to reduce health care utilization in general). The authors chose to use DID (difference in differences) and found a reduction in health care utilization (for outpatient and inpatient).

However, few issues:

1. Please do review the manuscript organization to align with PlosONE guidelines (see https://journals.plos.org/plosone/s/submission-guidelines)

- Note Abstract/Introduction/Materials and methods/Results/Discussion/Conclusions: All these pieces are present in the manuscript but they are not in that sequence

- For the readers of PlosONE it is too much to have such an Introduction + Background extension (6 pages and something more). Please just make on Background and do not put results or discuss your results on it. End this section with the aims.

2. The same issue of formatting with citations. Please see the guidelines. PLOS uses “Vancouver” style.

3. For the models, the authors state that they used the Poisson regression and a multiway approach to compute standard errors clustered both at the year-week and county level. Does this address potential overdispersion so common in count data?

4. Also, the authors indicate to have applied weighted regression by population size of a county. Why this weighting?

5. Please document the software used, and if possible share the code and the data used for this analysis. This is a quite pedagogic work.

6. I could not see the figure 3.

7. Table 1: These are mean and standard deviation. I wonder if Taiwan collects population and other demographic characteristics on weekly basis and at a county level. For example, the population pre- to post-outbreak changes. Why?

8. Tables 2, 3 and similar tables in the supplements present betas as they come from the software. I would recommend presenting in the main manuscript exponentiated version (which are relative changes) and their confidence intervals. And put the betas as of now in the supplements.

6. PLOS authors have the option to publish the peer review history of their article (what does this mean?). If published, this will include your full peer review and any attached files.

Reviewer #1: **Yes: **Sheikh Saifur Rahman Jony

Reviewer #2: **Yes: **Orvalho Augusto

---

## [Author Response · Author response to Decision Letter 0]

8 May 2022

We have prepared two rebuttal letters, labeled ”Response to Reviewer 1” and ”Response to Reviewer 2”.

---

## [Decision Letter · Decision Letter 1]

8 Jul 2022

Measuring Voluntary Responses in Healthcare Utilization During the COVID-19 Pandemic: Evidence from Taiwan

PONE-D-21-39908R1

Dear Dr. Yang,

We’re pleased to inform you that your manuscript has been judged scientifically suitable for publication and will be formally accepted for publication once it meets all outstanding technical requirements.

Kind regards,

M. Sohel Rahman, Ph.D.

Academic Editor

PLOS ONE

Additional Editor Comments (optional):

There are a couple of vey minor comments from Reviewer 2. I leave up to your discretion to consider those while you upload the final version.

Reviewers' comments:

Reviewer's Responses to Questions

**Comments to the Author**

1. If the authors have adequately addressed your comments raised in a previous round of review and you feel that this manuscript is now acceptable for publication, you may indicate that here to bypass the “Comments to the Author” section, enter your conflict of interest statement in the “Confidential to Editor” section, and submit your "Accept" recommendation.

Reviewer #1: All comments have been addressed

Reviewer #2: All comments have been addressed

2. Is the manuscript technically sound, and do the data support the conclusions?

Reviewer #1: Yes

Reviewer #2: Yes

3. Has the statistical analysis been performed appropriately and rigorously? 

Reviewer #1: Yes

Reviewer #2: Yes

4. Have the authors made all data underlying the findings in their manuscript fully available?

Reviewer #1: Yes

Reviewer #2: Yes

5. Is the manuscript presented in an intelligible fashion and written in standard English?

Reviewer #1: Yes

Reviewer #2: Yes

6. Review Comments to the Author

Reviewer #1: Authors has addressed all my comments properly. I think the paper can be accepted in current form.

Reviewer #2: This is a very good work and also pedagogical.

The Taiwan COVID-19 experience is unique, and the good data availability makes coupled with this analysis may teach the planet on how voluntary actions could largely change the course of an epidemic.

Few minor issues:

1. In the spirit of the pedagogy I would ask the authors, if they can, present in the appendix a snip of the year, week and the dummy indicators (for year, few for weeks). The formulas are good and clear. But remember not all people can follow this brilliant parametrization.

2. Figure 1. Please, add somewhere that this is for year 2020.

3. Lines 402 to 405 - OK for the possibility that measures lead to a "prevention effect". Isn't possible, as well, that people decided to hide they were sick?

7. PLOS authors have the option to publish the peer review history of their article (what does this mean?). If published, this will include your full peer review and any attached files.

Reviewer #1: No

Reviewer #2: **Yes: **Orvalho Augusto

---

## [Editor Report · Acceptance letter]

25 Aug 2022

PONE-D-21-39908R1 

Measuring Voluntary Responses in Healthcare Utilization During the COVID-19 Pandemic: Evidence from Taiwan 

Dear Dr. Yang:

I'm pleased to inform you that your manuscript has been deemed suitable for publication in PLOS ONE. Congratulations! Your manuscript is now with our production department. 

Kind regards, 

on behalf of

Dr. M. Sohel Rahman 

Academic Editor

PLOS ONE